# Traumatic Hip Dislocation in Pediatric Patients: Clinical Case Series and a Narrative Review of the Literature with an Emphasis on Primary and Long-Term Complications

**DOI:** 10.3390/children10010107

**Published:** 2023-01-04

**Authors:** Eetu N. Suominen, Antti J. Saarinen

**Affiliations:** Department of Orthopedics and Traumatology, Helsinki University Hospital, 00260 Helsinki, Finland

**Keywords:** hip dislocation, pediatric trauma, pediatric injury, avascular necrosis

## Abstract

Traumatic hip dislocation is a rare injury in pediatric populations. Dislocation may be associated with low-energy trauma, such as a minor fall. Traumatic hip dislocation is associated with severe complications, such as avascular necrosis of the femoral head. Timely diagnosis and reposition decrease the rate of complications. In this study we retrospectively assessed traumatic hip dislocations in pediatric patients during a 10-year timespan in a university hospital. There were eight cases of traumatic hip dislocations. All patients had a minimum follow-up of two years and were followed with MRI scans. One patient developed avascular necrosis during the follow-up which resolved conservatively. There were no other significant complications. In conclusion, traumatic hip dislocation is a rare injury which is associated with severe complications. Patients in our case series underwent a timely reposition. The complication rate was similar to previous reports.

## 1. Introduction

Traumatic hip dislocation is a rare injury among the pediatric population. Patients over 10 years of age typically require a major traumatic event for dislocation [1]. However, in younger children the dislocation may occur from a minor trauma as the traumatic energy required for hip dislocation increases with age [2,3,4]. Posterior dislocation is the most common type of injury, making up approximately 95% of traumatic dislocations [2]. Typical characteristics of posterior dislocation are adduction, flexion, internal rotation, and shortening of the dislocated limb [5]. Associated injuries increase with age and higher energy trauma. Associated acetabular and femoral fractures are uncommon in pediatric patients.

Closed reduction under moderate conscious anesthesia can be used in adolescent patients. Reduction is usually easy to achieve, but interfering fragments originating from the hip joint structures may prevent concentric reduction. In young children, a closed reduction in an operating room with readiness to convert to open reduction is recommendable [5]. If the operating room is unavailable, closed reduction should be performed in the emergency department. The successful reduction should be confirmed using fluoroscopy [6]. Widening of the injured joint space indicates unsuccessful reduction. If a congruent and stable hip joint is not achieved using the closed reduction, an open reduction is needed. After successful reduction, immobilization with a hip spica cast is recommended for children under 10 years. Protected weight bearing for 6–8 weeks is recommended for older patients [3].

Avascular necrosis (AVN) is the feared complication of hip dislocation. The only statistically proven risk factor is delay between dislocation and reduction. If the delay is greater than 6 h, the risk of avascular necrosis is increased 20-fold [2,3,6]. Reduction may happen spontaneously, especially in young children [3]. Magnetic resonance imaging (MRI) during the primary care is warranted for patients with suspected spontaneously reduction or with signs of unsuccessful reduction to assess soft tissue entrapments not visible on radiographs [7]. As AVN may develop up to two years after the injury, follow-up clinical surveillance and control MRIs are warranted.

In this study, we describe a patient series from a tertiary university hospital during a ten-year period. We also present a narrative literature review with an emphasis on primary complications and complications during extended follow-up after traumatic hip dislocation in pediatric patients.

## 2. Materials and Methods

Institutional review board permission was obtained for the study. A consecutive case series of acute, traumatic hip dislocations in skeletally immature patients between 1 January 2009 and 31 December 2019 treated in the pediatric orthopedic department of Turku University Hospital, Finland were reviewed retrospectively. A retrospective cohort of patients was collected using the International Classification of Diseases (ICD) code for hip dislocation (S73) to identify the cases from the patient charts of our center. Inclusion criteria were the correct diagnosis of traumatic hip dislocation, skeletally immaturity, minimum of two years of follow-up, and adequate clinical and radiographic information. Non-traumatic dislocations were excluded from the study. No patients with traumatic dislocations were excluded.

Patient characteristics and the etiology of the injury were collected from the patient charts. Approximate time from injury to reduction was assessed. Associated injuries and successful reduction were assessed from radiographs. MRIs taken during the follow-up were evaluated. Complications were assessed during the follow-up. The mechanism of injury was defined as high energy if it was because of a motor-vehicle accident or a fall from a height of more than 3 m (10 feet). A low-energy injury was defined as a typical play activity or fall from a height of less than 3 m (10 feet) [8].

## 3. Results

During the study period, eight patients treated for traumatic hip dislocation were identified. Four patients were male and four female. The mean age at the time of the injury was 11.7 years (range from 4.4 to 16 years). Three patients had a dislocation after a low energy trauma. Two of these patients were aged four years, and one patient was seven years at the time of the accident. Five patients had a hip dislocation after a high energy trauma. These patients were 14 to 16 years old. Six patients had posterior and two had anterior dislocations. The detailed descriptions of the patients and accidents are presented in Table 1.

Three patients over 10-years old had associated fractures. One patient had a Hill–Sachs lesion of the femoral head with loose extra-articular osteochondral fragment at the caput–collum border. This patient had an anterior dislocation of the femoral head. Two patients had acetabulum fractures as an associated injury. In one of these patients, an MRI showed a posterocranial fracture of the acetabulum with a 7 mm lateral dislocation between acetabulum and ramus. In another patient, a caudal acetabulum fracture was observed with a 25 mm × 13 mm sized dislocated fragment. These fractures were treated conservatively. None of the patients developed complications during the follow-up. There were no recurrent dislocations.

The mean time to reduction was approximately 3 h (range from 2 to 5 h). During the follow-ups, there were no radiological or clinical evidence of growth disturbances, heterotrophic ossification, early epiphysis closure, or post-traumatic arthritis. None of the patients had abnormal neurological or vascular findings in the lower extremity. None of the patients required surgical intervention for reduction or associated injuries.

The mean duration of the follow-up was 7.9 years (range from 4.1 to 9.8 years). One patient presented with bone edema and flattening of the caput indicating AVN at 4 weeks after the injury (Patient 7 in Table 1). Time to successful reduction in this patient was five hours. The patient was treated conservatively with a 6-month weight bearing prohibition. AVN was resolved at 6 months MRI control and the patient was asymptomatic in the clinical examination.

## 4. Discussion

Several studies in recent history have studied the epidemiology, mechanism of injury, clinical presentation, diagnosis, and treatment of the traumatic hip dislocation in a pediatric population. The findings of our study are consistent with the results of the previous research. Traumatic hip dislocation in the pediatric population is an uncommon injury that requires timely diagnosis and intervention. Due to low traumatic energy or untypical etiology, pediatric hip dislocation may go unnoticed [9,10]. Special caution is needed with unconscious patients in whom the ambulatory status is unknown.

Traumatic posterior hip dislocation in children often presents with the classic lower limb deformity. The hip is in flexion, adduction, and internal rotation. The involved limb appears shorter than the contralateral limb and the femoral head can be palpated posteriorly. Radiographs should be obtained without delay after adequate pain medication. The presence of another fracture or injury can divert attention from hip dislocation and delay the diagnosis. The dislocation of the femoral head can be anterior or posterior. Similar to adult patients, the majority of hip dislocations are posterior [11]. Bilateral dislocations have been reported in the literature but are exceptional [12]. A prompt closed reduction with the patient under suitable anesthesia and analgesia is considered to be appropriate treatment. Closed reduction is usually accomplished via traction in line with the deformity. In cases in which the diagnosis of hip dislocation is delayed, some studies suggest a short period of skeletal traction after the successful reduction until pain is improved. However, the advantage of this in order to achieve easier reduction has been compromised [10].

The traumatic energy required for a traumatic hip dislocation increases with age [1,5,6]. This was also seen in our patient sample as patients younger than 10 years had minor traumatic events in contrast to adolescents, who all had high-energy trauma. When compared to adult patients, associated fractures are rare in pediatric patients after traumatic hip dislocation. Large posterior wall fractures or fractures with articular surface dislocation require internal fixation to prevent hip instability.

Radiographic assessment by anteroposterior, oblique, and lateral imaging of the pelvis is crucial before and after reduction. Plain radiographs and CT scans may not adequately show acetabulum fractures in young children due to unossified bone. In these patients an MRI is warranted [7,13,14]. MRIs also provides information on soft-tissue injuries, including labral and capsular tears along with cartilage and muscular injuries [7]. Widening of the articular space in radiographs can be caused by entrapment of soft-tissue and requires imaging with an MRI. Archer et al. reported a patient with unrecognized physeal injury in whom the attempted closed reduction resulted in a hip fracture requiring open reduction and internal fixation [5]. When any uncertainties exist regarding hip joint congruency, a CT or MRI scan should be performed to identify intra-articular bone fragments or interposed soft tissue. Concerning the additional injuries connected with traumatic hip dislocation in pediatric patients, an acetabular fracture can be underestimated with radiographs and CT scans [13].

In our patients, no severe complications occurred during the follow-up. One 4.6-year-old patient presented with AVN after a posterior dislocation after landing with the hip on abduction and flexion on a bouncing castle. Dislocation was reduced after approximately five hours from the injury with a closed reposition in the operating room after which the patient was fitted with a spica cast. AVN was discovered in a routine MRI scan at four weeks after the injury. After the imaging finding, the patient was assigned to non-weight bearing for six months. AVN resolved during the follow-up at six months and patient returned to normal activities without difficulties.

Reduction in the dislocated joint restores the normal blood flow to the femoral head. The risk of osteonecrosis is substantial and may occur in up to one-third of dislocations, depending on the severity of injury. AVN of the femoral head is the most feared and frequent serious complication related to traumatic hip dislocation. Symptoms of AVN include painless limp, pain, and restricted movement. The incidence of AVN in children younger than 18 years is reported to be 3% to 15% after an isolated hip dislocation [15,16]. AVN is caused when the arterial perforation is disturbed. Precise pathological process of the AVN remains unclear. In a cadaver study, Chung demonstrated incomplete anastomotic and transepiphyseal vessels in skeletally immature patients [17]. Undeveloped anastomotic perforation exposes the femoral head to ischemia during the dislocation [2,17]. AVN may lead to severe disability and need for a hip replacement. Evidence at present indicates an association between late hip reduction and higher rate of osteonecrosis of the femoral head in all traumatic hip dislocations [18]. Hence, all traumatic hip dislocations should be reduced as soon as possible to decrease the rate of osteonecrosis of the femoral head. When a traumatic hip dislocation is associated with femoral epiphysiolysis, the risk increases to 100% [19,20]. Because of the risk of AVN, patients should be routinely followed with MRI scans. In our study, timely reduction was achieved in all the patients and the patients were adequately followed after the dislocation.

Primary neurologic complications associated with pediatric hip dislocations are rare, presumably due to the low traumatic energy typical to these injuries. The peroneal branch of the sciatic nerve is the most likely peripheral nerve to be injured. Sciatic nerve injuries are reported in the literature with an incidence rate of approximately 5% [21]. Nerve injuries associated with traumatic hip dislocations are usually a neurapraxias and symptoms are most often transitory. The peroneal nerves are most commonly affected. Pre- and post-reductive neurovascular status should always be examined. In our patient series, no neurologic complications occurred. Other uncommon complications of the traumatic hip dislocation in pediatric patients include recurrent dislocation, heterotrophic ossification, and development of coxa magna. Recurrent dislocation may be associated with a defect in the capsule or attenuation of the hip capsule without a tear. Acute redislocation, which occurs soon after reduction, implies that the reduction was not congruent and additional investigation with a CT scan is indicated with probable open exploration [11]. Proximal femoral head displacement (epiphyseolysis) during reduction in hip dislocation has been reported in the literature [19]. The possibility of physeal fracture should be kept in mind in traumatic hip dislocation occurring in children and adolescents with an open physis.

The development of the complications in traumatic hip dislocation can be delayed several years. Older studies before follow-up with MRIs have reported avascular necrosis after three years of the injury [22]. Patients should be subjected to long-term follow-up.

Neglected or untreated traumatic hip dislocations are rare. There are few reports on neglected traumatic dislocation. Although previous studies and case reports have presented a wide variety of treatment, the optimal management remains unclear. At a time-interval between injury and reduction procedure beyond 3–4 weeks, a reductional operation should be performed [16,23]. Even if the open reduction is delayed, it may prevent deformity, and will maintain the length of the lower extremity [24]. Regarding the surgical technique, it has been proposed that the surgeon should release the adductor longus, lengthen the psoas tendon, and insert a K-wire [16]. As mentioned, the traumatic hip dislocation in children is, in majority of the patients, an isolated injury, without a concurrent acetabular fracture. Therefore, the surgical technique should be chosen so that the operation does not further compromise the articular cartilage or the vascular structures of the femoral head.

No study to date has been able to define the type of rest required or the length of time for which it should be applied and there is no evidence that this affects the long-term outcome. Early gentle range of motion and patient mobilization should be instituted. Under the age of 10 years, immobilization with a spica cast for 4 weeks, along with suitable rehabilitation is particularly important for the healing of surrounding soft tissues to make the joint become stable [1]. Yuksel et al. suggest that the non-weight bearing interval of 4 weeks and partial weight bearing after 6 weeks are appropriate periods to let the soft tissue heal [6]. According to Sahin et al., neither the type of the treatment (post-reduction traction or bed rest) nor the time from injury to full-weight bearing influenced outcomes significantly [25].

The small sample size of our patient series prevented us from performing statistical analyses. Most prior reports on traumatic hip dislocations in pediatric patients are case reports or small case series. We report a modern single center case series with extensive follow-up with routinely taken MRI scans.

## 5. Conclusions

Traumatic hip dislocation is a rare injury in pediatric patients and is often associated with low-energy trauma. Adequate diagnosis and repositioning prevent complications, such as avascular necrosis of the femoral head. Radiographs should be obtained when a pediatric patient is unambulatory after a traumatic event, such as a fall. A shortened leg held in adduction and internal rotation should prompt the emergency physician to consider ordering radiographs and provide analgesia for the child. Primary MRI scans might be warranted to exclude associated injuries. During follow-up, MRI scans should be routinely taken to rule out avascular necrosis. The possibility of other associated injuries should be taken into account, especially in the patients with high-energy trauma. With appropriate management, most children with traumatic hip dislocation treated promptly with closed reduction will have an excellent outcome.

## Figures and Tables

**Table 1 children-10-00107-t001:** Clinical characteristics and etiology of the injuries.

ID	Age at Injury (Years)	Follow-Up (Years)	Approximate Time to Reduction (Hours)	Sex	Description of the Accident	Type of Dislocation	Associated Injuries
1	15.5	12.0	3.5	Male	Motocross injury, fell after jump and landed on the hip	Anterior	Hill–Sachs lesion of the femoral head, loose extra-articular osteochondral fragment at the caput–collum border
2	15.4	10.6	2.0	Female	Fell with a moped	Posterior	
3	14.9	10.1	5.0	Male	Downhill skiing injury, fell on the hip after a jump	Posterior	
4	7.3	8.7	2.3	Female	Fell on a trampoline	Anterior	
5	16.0	6.2	unknown	Male	Downhill skiing injury, fell on the hip after a jump	Posterior	Radiograph showed 6 mm fragment next to caput. MRI showed posterocranial acetabulum fracture. 7 mm lateral dislocation between acetabulum and ramus.
6	15.8	5.9	2.0	Male	Motocross bike collided with patient	Posterior	Caudal acetabulum fracture, 25 mm × 13 mm dislocated fragment.
7	4.6	5.7	3.0	Female	Landed with the hip on abduction on a bouncing castle	Posterior	
8	4.4	4.0	3.0	Female	Sledge collision with an adult	Posterior	

## Data Availability

Study data can be obtained from the corresponding author for a reasonable request.

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
