# Peer review of "Traumatic Hip Dislocation in Pediatric Patients: Clinical Case Series and a Narrative Review of the Literature with an Emphasis on Primary and Long-Term Complications"

_children, 2023, doi:10.3390/children10010107_

Round 1
Reviewer 1 Report
The manuscript is a retrospective study aimed to assessed traumatic hip dislocations in pediatric patients during a 10-year timespan from a university hospital. There were eight cases of traumatic hip dislocations. All patients had a minimum follow-up of two years and were followed with MRI scans. One patient developed avascular necrosis during the follow-up which resolved conservatively.
I read the article with interest, the title is well thought out and faithfully reflects the content of the study. The abstract is adequately developed, and it is useful to frame the characteristics of the study.
In the introduction, the characteristics of traumatic hip dislocation have been shortly described.
Nevertheless, it is a descriptive study with a sample of examined patients insufficient to perform a statistically significant evaluation. It also adds nothing to the international scientific panorama.
Author Response
We would like to thank the reviewer for taking the time to review our manuscript.
Comment 1: I read the article with interest, the title is well thought out and faithfully reflects the content of the study. The abstract is adequately developed, and it is useful to frame the characteristics of the study.
Response 1: We thank the reviewer the comments.
Comment 2: Nevertheless, it is a descriptive study with a sample of examined patients insufficient to perform a statistically significant evaluation.
Response 2: We agree with the reviewer that the sample of the patient is insufficient to perform statistical evaluation. This results from the fact that traumatic hip dislocation is a very rare injury. Therefore this, and majority of the previous research concerning this injury has been reported in the form of descriptive studies.
Comment 3: It also adds nothing to the international scientific panorama.
Response 3: We reported a modern case series with systematic follow-up. Our results are in line with prior modern studies. As most of reports are case reports or small case series, we believe that our sample provides useful information on the matter.
Reviewer 2 Report
Dear Sirs,
Your paper seems well structured but some aspects have to be addressed.
First of all, in which way Your work is different from other similar articles on this theme published in the last twenty years? Please explain it.
In the title You cited a literature review, but in the method section You should better define this aspect. Is this a narrative/comprehensive review? How did You do it? Please explain in the text.
Did You receive a ethical approval for Your work? Is it a retrospective case series? Please specify it.
The discussion section should be integrated. Which are the lapels of Your findings? Can we obtain useful evidence from this to improve rehabilitation and therapies for this disease? Just to has an example guide of what to do, consider this reference:
Notarnicola, A., Farì, G., Maccagnano, G., Riondino, A., Covelli, I., Bianchi, F. P., . . . Moretti, B. (2019). Teenagers’ perceptions of their scoliotic curves. an observational study of comparison between sports people and non- sports people. Muscles, Ligaments and Tendons Journal, 9(2), 225-235. doi:10.32098/mltj.02.2019.11
Best regards and good luck
Author Response
Comment 1: First of all, in which way Your work is different from other similar articles on this theme published in the last twenty years? Please explain it.
Response 1: We reported a modern case series with systematic follow-up protocol. Most prior studies are case reports or small case series. Our results are similar to prior studies. However, as most of the prior literature consist of small sample sizes we believe that further evidence is warranted.
Comment 2: In the title You cited a literature review, but in the method section You should better define this aspect. Is this a narrative/comprehensive review? How did You do it? Please explain in the text.
Response 2: In this study, we conducted a narrative review. Text added.
Comment 3: Did You receive a ethical approval for Your work? Is it a retrospective case series? Please specify it.
Response 3: This was a retrospective case series, this information has been added in Materials and methods section. No ethical approval from our Institution was required for this study. We were not in contact in patients and do not use identifying information.
Comment 4: The discussion section should be integrated. Which are the lapels of Your findings? Can we obtain useful evidence from this to improve rehabilitation and therapies for this disease?
Response 4: The Discussion has now been updated.
Reviewer 3 Report
Traumatic hip dislocation is an uncommon trauma in children traditionally associated with the high risk of complications. Despite of the classical descriptions in the textbooks and papers the more “fresh” data can change the practice in the light of the new data, the new practice and population or environmental changes. The authors aimed to describe a patient series during a ten-year period and present a literature review with an emphasis on complications in pediatric patients. A consecutive case series of acute, traumatic hip dislocations in 8 skeletally immature patients was analysed. The authors provided meticulous follow up of the patients focusing on the complications (including avascular necrosis). The general conclusion are consistent with the previously reported in the literature and basically support the necessity of the following the patients after traumatic hip dislocation (with the focus of the neurological problems like sciatic nerve damage and vascular complications needed MRI confirmation). Despite of the lack of the critically new data the paper gives an input to the general information regarding the problem and confirms the results of the previous studies.
Author Response
Comment: Traumatic hip dislocation is an uncommon trauma in children traditionally associated with the high risk of complications. Despite of the classical descriptions in the textbooks and papers the more “fresh” data can change the practice in the light of the new data, the new practice and population or environmental changes. The authors aimed to describe a patient series during a ten-year period and present a literature review with an emphasis on complications in pediatric patients. A consecutive case series of acute, traumatic hip dislocations in 8 skeletally immature patients was analyzed. The authors provided meticulous follow up of the patients focusing on the complications (including avascular necrosis). The general conclusion are consistent with the previously reported in the literature and basically support the necessity of the following the patients after traumatic hip dislocation (with the focus of the neurological problems like sciatic nerve damage and vascular complications needed MRI confirmation). Despite of the lack of the critically new data the paper gives an input to the general information regarding the problem and confirms the results of the previous studies.
Response: We thank the reviewer for taking time to review our manuscript and the comments.
Round 2
Reviewer 1 Report
Thank you for giving the necessary information, now the article is suitable for publication